# Preparation and Characterization of Calcium Carbonate Masterbatch–Alkali Soluble Polyester/Polyester Porous Fiber via Melt Spinning

**DOI:** 10.3390/ma17010160

**Published:** 2023-12-28

**Authors:** Yanjiao Zhao, Ruochen Song, Runan Pan, Meiling Zhang, Lifang Liu

**Affiliations:** College of Textiles, Donghua University, Shanghai 201620, China; stsmaoma@163.com (Y.Z.); songrc0509@163.com (R.S.); prnnan@163.com (R.P.)

**Keywords:** porous fiber, calcium carbonate particles, alkali soluble polyester (COPET), melt spinning, polyester

## Abstract

Porous fibers have gained significant attention for their lightweight and high porosity properties in applications such as insulation and filtration. However, the challenge remains in the development of cost-effective, high-performance, and industrially viable porous fibers. In this paper, porous fibers were fabricated through the melt spinning of an alkali soluble polyester (COPET)– CaCO_3_ masterbatch and PET slice. Controlled alkali and acid post-treatment techniques were employed to create porous structures within the fibers. The effects on the morphology, mechanical, thermodynamic, crystallinity, pore size, and thermal stability were investigated. The results indicate that the uniform dispersion of CaCO_3_ particles within the fiber matrix acts as nucleating agents during the granulation process, improving the thermal resistance and strength of the porous fiber. In addition, the porous fiber prepared by COPET/CaCO_3_ to PET with an 85/15 ratio and post-treated on 4% NaOH and 3% HCl exhibits a “spongy body” with uniformly small pores, favorable strength (2.71 cN/dtex), and elongation at break (47%).

## 1. Introduction

Porous fibers have garnered significant attention due to their wide range of applications, including insulation and filtration. Various methods have been employed for the preparation of porous fibers, including coaxial wet spinning [1], electrostatic spinning [2], microextrusion foaming [3], and melt-blow blending [4]. While these methods have demonstrated technical stability and the ability to replicate various pore structures, the highest porosity achieved in fiber-grade materials is currently limited to around 67% [5]. Additionally, carbonized preparation methods, although yielding higher porosity, are unsuitable for fiber-grade raw materials.

The fusion method has been explored for the preparation of porous fibers, with a focus on creating pores within the fiber wall. Two main approaches have been undertaken in previous studies. First, highly drawn fibers have induced the separation of the lamellar structure, resulting in micropores [6]. Second, particle blends or pore-making agents have been added to polymer raw materials, which are subsequently dissolved and removed after melt spinning to obtain porous fibers. Inorganic filaments containing Li^+^ and Ca^2+^ [7,8,9,10] have been commonly used as particle blends, with CaCO_3_ particles having advantages such as low cost, ease of filling, and high yield. For instance, a polyvinylidene fluoride (PVDF)/CaCO_3_/composite diluent was fused and blended to prepare a PVDF porous film, and the porosity was improved through acid treatment [11]. Polylactic acid (PLA) was modified by the fusion blending method of alginate and CaCO_3_, resulting in lower tensile strength and higher impact strength [12]. Additionally, the addition of nano-calcium carbonate increased the crystallinity and fracture strength of polypropylene/polyamide 66 [13].

Regarding filling CaCO_3_ to modify traditional fibers such as polyester and polypropylene, the technique is mature enough to meet the requirements of industrial production. However, this technology is now mostly used in the plastic industry, but in the field of textile application, it is still in the immature stage. In particular, there are few reports on the preparation of fiber grade by melt spinning by mixing calcium carbonate particles and alkali soluble polyester masterbatch granulation. A recent study focused on the preparation of a g-C_3_N_4_/CaCO_3_ composite catalyst mixed with PET/COPET powder, resulting in g-C_3_N_4_-based composite photocatalytic fiber prepared by melt spinning. However, this study mainly examined the effect of the catalyst, with limited investigations on the performance of fiber-grade products [14].

Considering the increasing demand for lightweight and thermally functional materials in the market, this study seeks to address the issue by modifying traditional polyester composite materials. The modification involves granulating CaCO_3_ particles with an alkali soluble COPET polyester masterbatch and blending them with a conventional polyester masterbatch after the melting process. The effects of COPET-CaCO_3_ content, alkali treatment, and acid treatment on the resulting porous fibers’ pore size structure, thermal properties, and mechanical properties were explored. By systematically examining these parameters, this study aims to provide a comprehensive understanding of the relationship between the processing conditions and the resulting fiber characteristics. The findings will provide valuable insights into the development of porous fiber materials with enhanced functionality, addressing the growing demand for lightweight and thermally efficient materials in industries such as construction, transportation, energy storage, and textiles.

## 2. Experiment

### 2.1. Materials and Instruments

COPET/CaCO_3_ masterbatch (abbreviated as CTCA masterbatch) (self-made, COPET/CaCO_3_/modified polyester wax mju dispersant/68.5/30/1.5, [η] = 0.332 dL/g, filter press value of 5.01 MPa; among them, COPET made by adding modified components to ordinary PET synthesis, including sodium bis(2-hydroxyethyl) 5-sulfoisophthalate (SIPE), ethylene glycol (EG), and other components, and SIPE monomer mainly used in the synthesis of polyester, which is beneficial to improve the solubility and dyeing property of polyester); PET slice ([η] = 0.71 dL/g, Tm = 256 °C), produced by Yichang Zhongying Technology Development Co., Ltd. (Hubei;, China); sodium hydroxide (96%), purchased from Shanghai Titan Technology Co., Ltd. (Shanghai, China); hydrochloric acid (36–38%) provided by Sinopharm Group Chemical Reagent Co., Ltd. (Shanghai, China); penetrant surfactant cetyltrimethylammonium bromide (CTAB, 96%), Tianjin Damao Chemical Reagent Factory (Tianjin, China); SHR High Speed Mixer, 20 L, Zhangjiagang Mingsu Machinery Co., Ltd. (Suzhou, China) (40 preoriented yarn “POY”); single-screw spinning machine, screw diameter of 40 mm, 48 f, length–diameter ratio of spinneret hole at 2, diameter of single spinneret hole at 0.6 mm, ratio of hollow yarn to solid yarn at 24.41%, winding speed of 2800 r/min, supplied by Suzhou Baolidi Material Technology Co., Ltd. (Suzhou, China); constant temperature digital display water bath, DK-S28, Shanghai Xiyuan Experimental Instrument Co., Ltd. (Shanghai, China); vacuum drying oven, DZF-6020A, Shanghai Yiheng Scientific Instrument Co., Ltd. (Shanghai, China); electronic balance, AL104-IC, Mettler Toledo Instrument Co., Ltd. (Shanghai, China).

### 2.2. Preparation of Porous Fibers

The fusion spinning method and post-treatment process were used to prepare porous fibers, and the preparation process is shown in Figure 1.

#### 2.2.1. Melt Spinning

First, set up gradient heating and drying for the CTCA masterbatch. The CTCA masterbatch was first dried in a vacuum oven at 80 °C for 4 h, and then dried at 110 °C for 12 h. A PET slice was dried in a vacuum drying oven at 110 °C for 5 h. After thorough drying, the self-made COPET/CaCO_3_ masterbatch (denoted as CTCA masterbatch) and PET slice were blended at different ratios (95/5 as A, 85/15 as B, and 80/20 as C), were mixed in a high-speed mixer at 80 °C, 400 V voltage, 700 r/min for 4 consecutive times, each time for 8 min. The resulting mixture was subjected to melt spinning using a 40 POY single-screw spinning machine equipped with a spinneret consisting of 48 holes with a diameter of 0.6 mm. The die head temperature was maintained between 278 and 288 °C, while the spinning temperature ranged from 265 to 295 °C across intervals. The metering pump frequency was set at 22–23 Hz, and the winding speed was maintained at 2800 r/min. These parameters were carefully controlled to ensure precise and accurate fabrication of the porous fibers.

#### 2.2.2. Postprocessing

Alkali treatment. The medium spinning yarns A, B, and C discussed in Section 2.2.1 underwent alkali treatment at 100 °C for 30 min, with a bath ratio of 1:15 and a 0.2% cetyltrimethylammonium bromide (CTAB) penetrant. The NaOH concentration for each sample was varied as mass fractions of 0%, 1%, 2%, 3%, 4%, 5%, and 6%. Following treatment, the samples were washed five times with deionized water, followed by five washes with anhydrous ethanol. Subsequently, they were dried in a vacuum drying oven at 100 °C for 2 h. The samples were labeled as A-1 and so on, represented the treated samples of sample A at different NaOH concentrations.

Acid treatment. The samples obtained from the alkali treatment in the previous step were further subjected to hydrochloric acid treatment at room temperature for 1 h. The acid concentrations used for each sample were 1, 2, 3, and 4 mol/L. Following the acid treatment, the samples underwent the same washing and drying procedures as described in the alkali treatment. The resulting samples were labeled as A-1-1, A-1-2, and so forth, signifying PET/CACT/95/5 after specific alkali and acid concentration treatments.

### 2.3. Characterization

#### 2.3.1. Base (Acid) Decrement

①Alkali treatment alkali decrement

The samples before and after alkali reduction treatment were dried in a vacuum drying oven. After drying and balancing in a dryer for 24 h, the dry weight before alkali reduction treatment and after alkali reduction treatment was weighed. The alkali decrement formula is calculated as follows [15,16,17]:(1)$$W=\frac{m_{0\;}-{\;m}_{1}}{m_{0}}$$

W—the decrement rate, %;

m_0_—dry weight before treatment, g;

m_1_—dry weight after treatment, g.

②Acid treatment acid decrement

The acid treatment method employed was similar to the one used in the alkali treatment and alkali decrement experiment (described in ①).

#### 2.3.2. Strength and Linear Density

The determination of strength and linear density was conducted in accordance with the guidelines specified in the reference [16] GB/T 14335-2008, “Test Method for Linear Density of Chemical Fiber Staple Fibers”, and [17] GB/T 14337-2022, “Test Method for Tensile Properties of Chemical Fiber Staple Fibers”. For this purpose, the FAVIMAT+ automatic single-fiber strength meter (Textechno Herbert Stein GmbH & Co. KG, Mönchengladbach, Germany) was utilized. The instrument provided a test range of 0 cN–1200 cN with a resolution of 0.0001 cN. The maximum test length was set at 100 mm, while the elongation resolution was maintained at 0.0001 mm. The sample was reduced to wet equilibrium under standard atmospheric conditions.

#### 2.3.3. Tungsten Filament Scanning Electron Microscope

The transverse and longitudinal morphologies of samples were observed using an EVO 10 Zeiss tungsten filament scanning electron microscope (Carl Zeiss Optics Co., Ltd., Shanghai, China), and the accelerated voltage was 0.2–0 kV. During sample preparation, the cross section was cut by a Y172 fiber slicer according to the reference [18] GB/T 10685-2007, “Wool Fiber Diameter Test Method Projection Microscope Method”. The longitudinal section was subjected to the liquid nitrogen embritching method. The sample was frozen in liquid nitrogen for 20 min, and then embritched with scissors frozen in liquid nitrogen. The sample was vertically adhered to the electron microscope table, and the sample was sprayed with platinum for 150 s at a current of 15 mA before testing.

#### 2.3.4. Specific Surface Area and Aperture Distribution

A BET ASAP 2460 multistation extended automatic specific surface and porosity analyzer (McMuriatic Instrument Co., Ltd., Shanghai, China) was used to test the gas adsorption isotherm of the sample to be tested according to the static adsorption equilibrium volume method. The untreated and treated fibers were randomly cut to 0.1 g each for testing. Combined with BET and BJH model principles, the specific surface area and pore size distribution were analyzed, respectively. The test conditions were N_2_ adsorption, the analysis tank temperature was −195.85 °C, the low pressure dose was 6.0 cm^3^/g, the equilibrium interval was 15 s, the degassation temperature was 100 °C, and the degassation time was 6 h.

#### 2.3.5. DSC and TG

An Elmer STA8000 synchronous thermal analyzer (PerkinElmer Co., Ltd., Waltham, MA, USA) was utilized to perform thermogravimetric analysis (TGA) and differential scanning calorimetry (DSC) tests according to the standards the reference [19] GB/T 27661-2011 and [20] GB/T 19466.2-2004. TGA involved a 6 mg sample mass, with a temperature rise rate of 20 °C/min under a nitrogen atmosphere (nitrogen rate of 50 mL/min) up to 800 °C. DSC included a temperature rise to 300 °C at 20 °C/min, followed by a 5 min hold, a rapid cool to room temperature, and a subsequent rise to 300 °C at 20 °C/min to eliminate thermal history. These procedures were conducted to meet the specific requirements outlined in the aforementioned standards.

#### 2.3.6. Fourier Infrared Spectrum

An Elmer Frontier tester (PerkinElmer Co., Ltd., Waltham, MA, USA) was used to analyze the composition, and the composition and structure of the sample were analyzed according to the photoacoustic spectrometry method specified in the reference [21] GB/T 6040-2019, “General Rules for Infrared Spectral Analysis Methods”. The wavelength scanning range was 350–8300 cm^−1^, the spectral resolution was better than 0.4 cm^−1^, and the number accuracy was as follows: better than 0.008 cm^−1^ SNR: 200,000:1 (1 min test, 4 cm^−1^ resolution, DTGS detector, RMS).

#### 2.3.7. X-ray Diffraction

The crystal structure of the samples was obtained by an ADS D8 advance X-ray diffractometer (Bruker Spectral Instrument Company, Billerica, MA, USA). The Cu target was a standard optical tube with a voltage of 40 kV, a current of 25 mA, a step width of 0.02°, a scanning speed of 0.15 s/step, and a scanning range of 10°–90°.

## 3. Results and Discussion

### 3.1. Base (Acid) Decrement

#### 3.1.1. Alkali Decrement

Figure 2 illustrates the relationship between alkali decrement and alkali concentration. It can be observed that, under constant temperature and time conditions, the alkali reduction of polyester POY yarn exhibits a nearly linear trend with increasing NaOH concentration. This behavior can be attributed to the presence of ester bonds on the PET molecular chain, which undergo irreversible hydrolysis upon reaction with NaOH, leading to partial degradation or even chain fracture. As the NaOH concentration rises, a greater number of hydroxyl groups (-OH) are available, thereby promoting the hydrolysis reaction [22,23].

CTAB, a series of quaternary ammonium surfactants, plays a significant role in reducing the interfacial tension between PET and NaOH aqueous solution. Thus, the NaOH solution is more easily immersed in the fiber interior, which is conducive to hydrolysis reaction. Moreover, an increase in the quantity of CTCA masterbatch results in a corresponding increase in alkali reduction. This can be attributed to the presence of SIPE in COPET within the CTCA masterbatch. The -SO_3_Na group in SIPE acts as a polar entity with electron absorption capabilities, enabling it to absorb more hydroxyl groups and intensify the hydrolysis reaction. Additionally, the mesostructure of SIPE disrupts the regularity of copolymerization molecules and diminishes their crystallization ability. Consequently, the NaOH solution penetrates the molecular chain, providing favorable conditions for the hydrolysis reaction [24].

#### 3.1.2. Acid Decrement

Figure 3 presents the relationship between acid reduction and acid concentration. At a constant NaOH base concentration, the acid loss increases with higher HCl concentrations. Similarly, when the HCl concentration and the amount of the CTCA masterbatch remain constant, the acid decrement increases as the NaOH base concentration rises. Furthermore, with a constant HCl concentration and NaOH base concentration, the addition of the CTCA masterbatch leads to an increase in acid reduction. This phenomenon can be attributed to the chemical reaction between CaCO_3_ particles present in the POY fiber, formed after HCl treatment, and the alkali solution. The reaction generates CO_2_ gas, which is released and causes a reduction in fiber weight. The intensity of this chemical reaction and the subsequent weight reduction are influenced by the concentration of HCl and the quantity of the CTCA masterbatch added. Higher HCl concentrations and increased amounts of the CTCA masterbatch result in more pronounced chemical reactions and a greater reduction in fiber weight. Moreover, a higher NaOH base concentration creates more surface holes on the fiber after alkali treatment. These surface holes expose a larger number of CaCO_3_ particles, facilitating their interaction with HCl and enhancing the reaction efficiency. As a consequence, the acid reduction increases with higher NaOH base concentrations.

### 3.2. Morphological Structure

#### 3.2.1. Scanning Electron Microscopy after Alkali Treatment

The surface morphology SEM images before and after fibronine treatment are shown in Figure 4. Untreated POY fibers with different CTCA contents exhibited smooth and cylindrical transverse and longitudinal sectional structures. CaCO_3_ particles were found to be evenly dispersed on both the cross section and surface, particularly on the longitudinal surface (highlighted by the yellow box in Figure 5). In the case of sample A, an increase in alkali concentration resulted in a slight increase in the number of holes observed in the longitudinal section, while the cross section experienced minimal changes. This correspondence between the increase in alkali concentration and the presence of additional holes supports the findings obtained from the alkali reduction analysis.

For samples B and C, with the alkali concentration kept constant, an increase in CTCA masterbatch content led to a notable increase in hole formation in both the transverse and longitudinal sections. This phenomenon can be attributed to the higher CTCA masterbatch content, which corresponds to an increased proportion of COPET containing alkali-soluble components. The presence of a modified link with sodium isophenate sulfonate in the COPET macromolecular chain introduces a sulfonic acid group that acts as an electron-absorbing moiety. This interaction reduces the electron cloud density on the carbonyl carbon, intensifying the electrostatic force between the carbonyl carbon and -OH. Consequently, the susceptibility of the carbonyl carbon to hydrolysis reactions is enhanced, resulting in the formation of holes on the fiber’s surface [25,26], as shown in Figure 6. Furthermore, the increased exposure of CaCO_3_ particles facilitates subsequent acid treatment. These observations align with the findings obtained from the alkali reduction analysis conducted on samples B and C.

For sample B, at an alkali concentration of 4%, numerous and uniform-sized pores were observed, resembling a “sponge” state. At higher alkali concentrations (5–6%), more pore structures appeared, with a tendency to form larger pores, potentially affecting fiber mechanical properties. For sample C, similar trends were observed at lower alkali concentrations. However, at higher concentrations (≥5%), the fiber sections collapsed, especially with the formation of large holes, significantly impacting fiber performance. Based on the findings, sample B demonstrated better pore morphology and strength properties at alkali concentrations of 4–6%, while sample C performed optimally at alkali concentrations of 2–4%.

#### 3.2.2. Scanning Electron Microscopy after Acid Treatment

To obtain porous fiber with uniform pore size and high porosity, we further treated B-4, B-5, B-6, C-2, C-3, and C-4 with acid. SEM images for corresponding samples are shown in Figure 7.

It was observed that increasing HCl concentration resulted in the gradual formation of a distinctive “honeycomb” and “cavernous” structure in the cross section of the fibers. This structural change was accompanied by the enlargement of pores and the appearance of cross-links between the pores and the longitudinal section morphology. The observed phenomenon can be attributed to the reaction between HCl and CaCO_3_ particles, leading to the generation and subsequent escape of gas, which leaves behind holes on the fiber surface [27]. The intensity of this reaction and the resulting pore formation increased with higher HCl concentrations. However, it should be noted that excessively high HCl concentrations led to the collapse of both transverse and longitudinal sections of the fibers, making it impossible to observe the complete pore structure. This collapse phenomenon has implications for the overall performance of the fibers and should be considered in practical applications.

Furthermore, the effect of NaOH and HCl concentrations on the fiber structure was investigated while keeping the CTCA masterbatch content constant. The results indicated that increasing both NaOH and HCl concentrations led to an increase in the number of holes observed in the transverse and longitudinal sections of the fibers. Notably, the holes observed in the cross section after acid treatment were more pronounced compared with those in the longitudinal section. Optimal fiber morphology was observed at a NaOH concentration of 4% and a HCl concentration of 3%, where the transverse and longitudinal sections exhibited a favorable structure. However, a further increase in NaOH concentration to 6% and HCl concentration to 3% resulted in a collapse of the fiber cross-section morphology.

### 3.3. Chemical Structure and Crystalline Property

Fourier infrared spectrum tests were conducted on fibers before and after treatment (Figure 8a). The spectrograms of all tested samples exhibited characteristic peaks consistent with PET spectra. Specifically, peaks at 1716, 1242, 1092, and 724 cm^−1^ corresponded to the stretching vibrations of the ester group’s C=O, ester-C(O)O stretching vibrations, benzene ring’s 1-stretching vibrations, and benzene ring’s 1-stretching vibrations, respectively. Flexural vibrations caused the presence of two substituted carbonyl groups outside the plane of the benzene ring and vibrations at the four-site substitution [14,28,29,30].

In the untreated samples, an increase in CTCA masterbatch content resulted in a significant increase in the peaks of infrared spectra at 1716, 1242, 1092, 724, and 1437.3 cm^−1^, indicating enhanced absorption representing the structure. Particularly, the peak around 1437.3 cm^−1^, corresponding to the main absorption peak generated by C-O vibration in CaCO_3_ [31], significantly increased, suggesting successful filling of the fiber by CaCO_3_ particles in the CTCA masterbatch. Upon alkali treatment, the peak in the range of 1600 to 1200 cm^−1^ exhibited slight changes, reflecting structural modifications resulting from hydrolysis reactions in the PET [32].

Figure 8b displays the X-ray diffraction (XRD) pattern of the samples. The diffraction peaks observed at 23°, 29.35°, 35.85°, 39°, and 48° for samples A, B, and C correspond to the standard calcite atlas of CaCO_3_, indicating the presence of calcite structures belonging to the cubic crystal system [33]. This confirms the successful loading of CaCO_3_ particles from the CTCA masterbatch into the fiber. Other samples also exhibit a characteristic peak at 29.35°, albeit with reduced intensity. Notably, the alkali-treated sample exhibits weaker diffraction peak strength compared with the untreated sample, but stronger than the acid-treated sample. This can be attributed to the removal of CaCO_3_ particles from the fiber surface during alkali treatment, causing them to burst out [34]. Acid treatment, on the other hand, involves direct chemical reactions with CaCO_3_ particles, although some particles may remain.

### 3.4. Thermal Analysis

Various parameters, including melting temperature (T_m_), cold crystallization temperature (T_c_), melting enthalpy (ΔHm), and crystallinity (Xc), were determined via DSC (Figure 9a, Table 1). The relationship between T_m_ and the amount of CaCO_3_ particles in the CTCA masterbatch is not significant due to the presence of COPET as the granulation substrate and the inorganic nature of CaCO_3_ particles. With increasing incorporation of CaCO_3_ particles, T_m_ of the sample shifts from a higher to a lower temperature. The addition of the CTCA masterbatch slightly enhances crystallinity, indicating the nucleating role of CaCO_3_ particles during granulation. However, when the amount of the CTCA masterbatch (and CaCO_3_ particles) remains constant, alkali and acid treatment decreases crystallinity by causing surface holes and compromising crystal integrity. DSC analysis shows that alkali and acid treatment has almost no effect on Tm but reduces crystallinity [35,36].

The thermal stability analysis of the samples in Figure 9b reveals a consistent thermal degradation behavior among all samples. The thermal decomposition temperature, approximately 390 °C, aligns with the thermal decomposition temperature of pure PET [14,37]. The thermal decomposition of the sample is divided into two stages. The first stage is about 30–600 °C, which is mainly COPET/PET resin matrix decomposition. The second stage is about 600–800 °C, which is the decomposition of CaCO_3_ inorganic particles [38,39]. In the first stage, the thermal decomposition temperature is about 390 °C, which is basically close to the thermal decomposition temperature of pure PET polyester [29,30]. For both untreated and alkali-treated samples, with the increase in CTCA masterbatch content, the thermal decomposition temperature of the samples first increased and then decreased. For the acid-treated samples, the thermal decomposition temperature decreased with the increase in CTCA masterbatch content. In the second stage, for untreated and alkali-treated samples, the carbon residue increased with the increase in CTCA masterbatch content. After acid treatment, with the increase in CTCA masterbatch content, the carbon residue increased first and then decreased. This is because the content of the CTCA masterbatch increases, the content of CaCO_3_ inorganic particles in the fiber increases, and the CaCO_3_ particles start to decompose at about 600–800 °C. Therefore, the addition of CaCO_3_ at this stage can increase the thermal decomposition temperature of the fiber, and also increase the quality of carbon residue. When treated with alkali, OH- reacts with COPET and PET on the fiber surface step by step, and finally forms pores on the fiber surface and inside. When the pore volume is small, it is helpful to increase the thermal decomposition temperature and carbon residue. However, when there are too many pores, CaCO_3_ particles adsorbed on the fiber will be detached from the fiber, and the fiber matrix itself will be hydrolyzed and consumed, so the thermal decomposition temperature and carbon residue will decrease. When the acid solution reacts with CaCO_3_ inorganic particles at the beginning, the reaction rate is slow. With the increase in CaCO_3_ particles, the amount of carbon residue increases, and with the increase in CTCA masterbatch content, the reaction rate with acid is accelerated, the consumption rate is greater than the addition rate of CaCO_3_ particles, and the gas generated by the reaction dissolves into pores, resulting in more pores on the fiber surface. The probability of CaCO_3_ particles separating from fiber is further increased, so the thermal decomposition temperature of fiber and the amount of carbon residue are reduced.

### 3.5. Strength and Linear Density

The preset yarn fineness is set as 200D/48f (that is, the fineness of a single filament is about 4.625 dtex), and the POY fiber obtained by spinning is directly used for post-treatment process.

#### 3.5.1. Strength and Density after Alkali Treatment

Table 2 analysis revealed that the linear density of untreated fibers showed a slight increase with increasing CTCA masterbatch addition, although the change was not significant. For the same amount of the CTCA masterbatch, the linear density of alkali-treated POY fiber exhibited an initial increase, followed by a decrease with increasing NaOH concentration, attributable to the preoriented nature of POY fiber. At 100 °C alkali treatment, enhanced mobility of both large and small molecules occurred. This increased macromolecular disorder and disorientation. The increased mobility of small molecules favored their penetration into the fiber, leading to large molecule degradation, fracture of the amorphous region, reduced shrinkage, shorter length, and coarser fiber [24]. Hydrolysis primarily transpired in the amorphous or crystalline region edges. Higher NaOH concentrations intensified hydrolysis, reducing the remaining amorphous region, diminishing shrinkage energy, and resulting in decreased fiber shrinkage.

The fiber fracture strength decreased with increasing CTCA masterbatch addition, and the same trend persisted with increased NaOH concentration for a given CTCA masterbatch level. Elevated CTCA masterbatch content raised inorganic particle content, resulting in increased brittleness, decreased fracture strength, and even exceptionally low fiber strength that failed to meet flocculation requirements [40]. When the CTCA masterbatch quantity remained constant, higher NaOH concentrations led to an augmented negative charge on the fiber surface, facilitating greater hydroxyl group adsorption. This enhanced hydrolysis reaction, forming surface pores and thinning the fiber, creating more stress concentration points, consequently causing a continuous decline in breaking strength [41,42].

#### 3.5.2. Strength and Density of Monofilament after Acid Treatment

The test data of fiber monofilament strength and linear density after acid treatment are shown in Table 3.

Under constant NaOH concentration and CTCA masterbatch addition, increasing HCL concentration leads to a decrease in both linear density and fracture strength. Similarly, when the CTCA masterbatch addition remains constant, increasing NaOH and HCL concentrations also result in decreased linear density and fracture strength (Table 3). This outcome can be attributed to the fixed total amount of CaCO_3_ particles and the fixed surface exposure on the fibers. With higher HCL concentration, the chemical reaction rate with CaCO_3_ particles intensifies, causing the formation of voids on the fiber surface due to the release of CO_2_ gas. Consequently, the linear density of the fiber reduces. Simultaneously, the fiber structure, responsible for providing support, undergoes damage, leading to a decrease in fracture strength until the reaction of CaCO_3_ particles is complete.

With constant NaOH and HCL concentrations, an increase in CTCA masterbatch addition initially raises linear density, followed by a subsequent decline, while fracture strength continues to decrease. The initial increase in linear density can be attributed to fiber shrinkage during high temperature treatment, resulting in fiber length reduction. The decrease in fracture strength and linear density can be attributed to the brittleness induced by higher CTCA masterbatch addition, resulting in decreased fiber strength, reduced fiber shrinkage, coarsening of fibers, and ultimately reduced fracture strength.

Based on the analysis of the data from Table 1 and Table 2, it can be concluded that samples A and B exhibit fracture strength properties that meet the requirements for the usage of floc fibers when the alkali concentration ranges from 1% to 6%, and the acid concentration ranges from 1% to 4%. For sample C, fracture strength properties can meet the requirements when treated with alkali concentration ranging from 1% to 4% and acid concentration ranging from 1% to 3%.

### 3.6. Porous Structure

The N_2_ adsorption–desorption isotherm diagram in Figure 10 was subjected to analysis. According to the systematic nomenclature (IUPAC) classification, samples from series A, B, and C exhibit type IV isotherms with H1 hysteresis loops [43,44], indicating the presence of fractured pores in these samples (Figure 11). Furthermore, the N_2_ adsorption–desorption curves of the samples show a nonclosed and cross-shaped pattern. This is likely due to the small specific surface area of the samples, which leads to poor closure behavior and fluctuations in the adsorption and desorption values, resulting in the intersection of the curves.

As observed in Figure 10, the adsorption capacity of series A, B, and C samples increases gradually in the range of P/P_0_ = 0.0 to 0.7, with N_2_ molecules being adsorbed in single to multiple layers on the inner surface of the mesoporous pores. When P/P_0_ = 0.7 to 1.0, a sudden increase in adsorption capacity is observed. The pore homogeneity can be evaluated via pore diameter and the width of fiber. Two distinct steps are observed in the ranges of P/P_0_ = 0.7 to 0.9 and P/P_0_ = 0.9 to 1.0, indicating the pore size distributions within two ranges. The pore sizes of fibers are concentrated around 40 and 50 nm. In the range of P/P_0_ = 0.9 to 1.0, an evident tailing phenomenon can be observed, which can be attributed to the presence of larger mesopores within the samples [45,46]. By comparing the samples from series A, B, and C, it is evident that the width of the hysteresis loops in the N_2_ adsorption–desorption curves of series B samples is consistently similar, indicating a higher aperture uniformity. The pore uniformity of series C is lower than that of series B but better than that of series A.

Table 4 provides specific surface area, pore volume, and pore size data for the prepared samples. In line with the N_2_ adsorption–desorption isotherm analysis, an increase in CTCA masterbatch content results in higher specific surface area and average pore size, with minimal impact on pore volume. This can be attributed to the presence of inorganic CaCO_3_ particles within the fiber matrix. Alkali and acid treatment of samples A and B initially increases specific surface area, pore volume, and average pore size, consistent with the direct pore distribution trend observed. However, further treatment leads to a decrease in these parameters. High-temperature treatment induces fiber shrinkage, resulting in increased fiber diameter, specific surface area, and pore volume. Additionally, alkali and acid reactions decompose the fiber’s amorphous region and dissolve CaCO_3_ particles, resulting in reduced fiber diameter, specific surface area, and pore volume. Exceeding the addition of fiber-grade powder with the CTCA masterbatch in the C-series samples accelerates the reaction rate between fiber and alkali/acid. As a consequence, alkali and acid treatment decreases specific surface area, pore volume, and average pore diameter, particularly in acid-treated samples, in line with the direct pore distribution observed.

## 4. Conclusions

In summary, this study presents a significant advancement in the development of cost-effective and high-performance porous fibers with lightweight and high porosity properties. A COPET-CaCO_3_ masterbatch and PET sliced fiber were prepared by melt spinning. Combined with controlled alkali and acid post-treatment techniques, a breakthrough in morphology, mechanical properties, thermodynamic behavior, crystallinity, pore size, and thermal stability of the porous fibers was achieved. The uniform dispersion of CaCO_3_ particles within the fiber matrix as nucleating agents demonstrated improved thermal resistance and strength of the resulting porous fibers. Notably, the composition of COPET/CaCO_3_ to PET (85/15) and post-treatment with 4% NaOH and 3% HCl yielded a distinct “spongy body” structure with uniformly small pores, exhibiting exceptional mechanical properties with a strength of 2.71 cN/dtex and an elongation at break of 47%. This study provides valuable insights and paves the way for the practical implementation of these porous fibers in diverse industrial applications, offering a promising solution for lightweight and high-performance materials.

## Figures and Tables

**Figure 1 materials-17-00160-f001:**
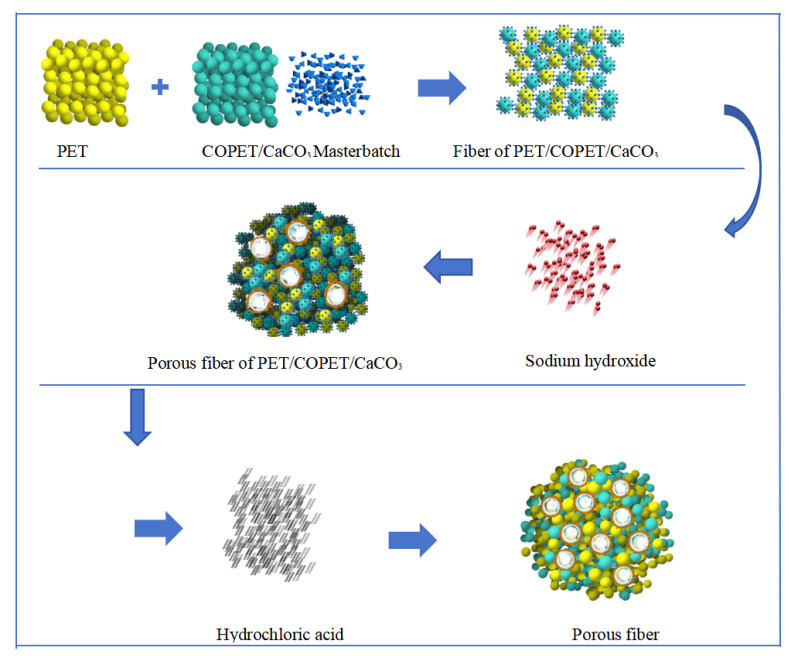
Fabrication process of CaCO_3_-COPET/PET porous fiber.

**Figure 2 materials-17-00160-f002:**
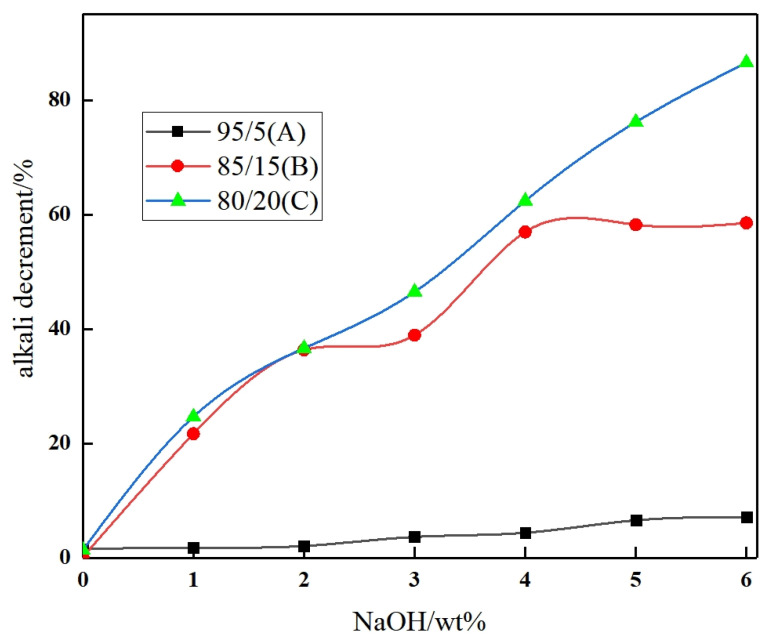
Relationship between alkali decrement and alkali concentrations of porous fiber.

**Figure 3 materials-17-00160-f003:**
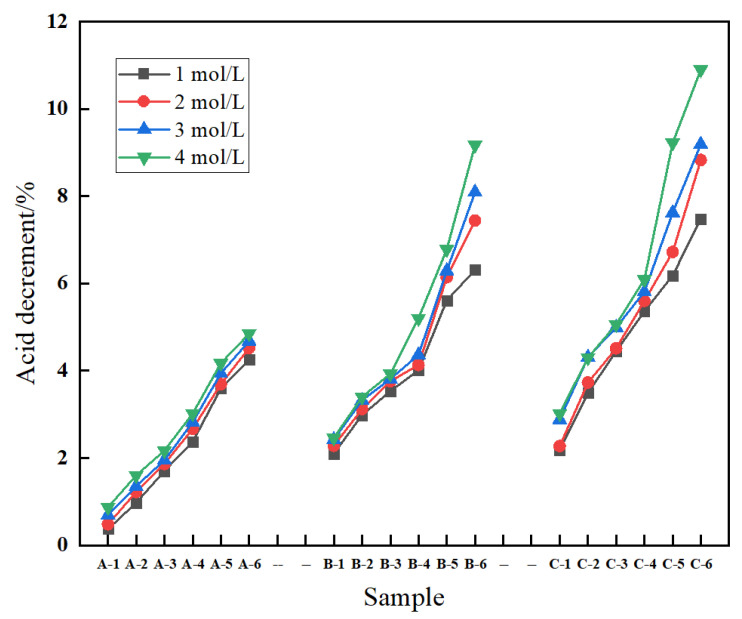
Relationship between acid decrement and acid concentrations of porous fiber.

**Figure 4 materials-17-00160-f004:**
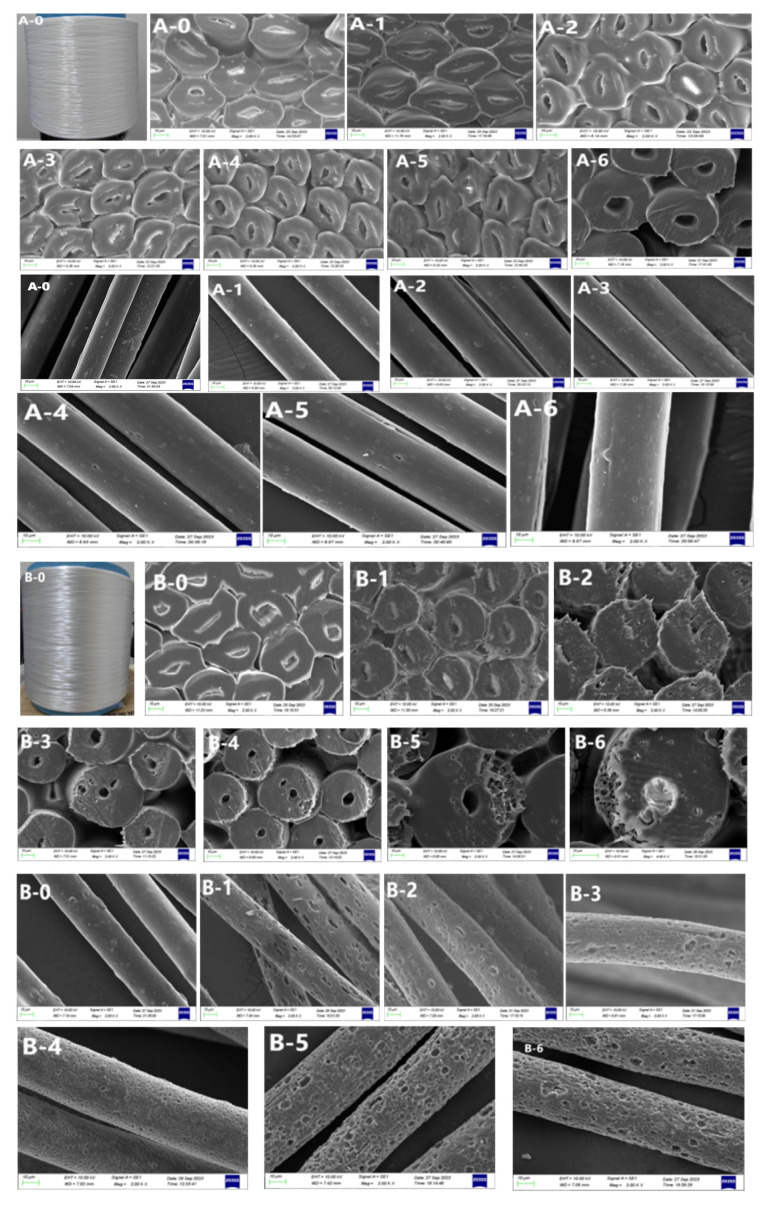
SEM images of POY fibers before and after alkali treatment: (A-0~A-6) is the transverse and longitudinal sectional morphology of sample A before and after alkali treatment; (B-0~B-6) is the transverse and longitudinal sectional morphology of sample B before and after alkali treatment; and (C-0~C-6) is the transverse and longitudinal sectional morphology of sample C before and after alkali treatment. Among them, A-0~A-6 appearing for the first time represents the fiber cross section topography, and A-0~A-6 appearing for the second time represents the fiber longitudinal section topography.

**Figure 5 materials-17-00160-f005:**
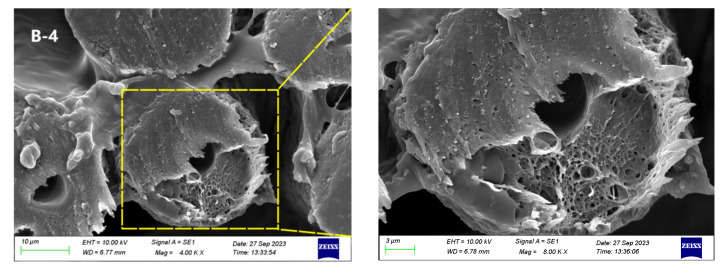
The “sponge” morphology of sample B-4 after alkali treatment.

**Figure 6 materials-17-00160-f006:**
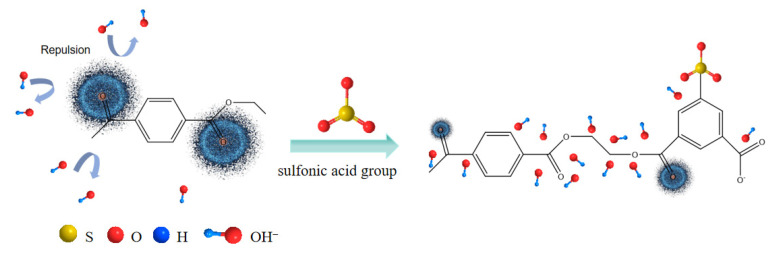
The electron cloud density of carbonyl carbon was reduced by introducing the sulfonic acid group.

**Figure 7 materials-17-00160-f007:**
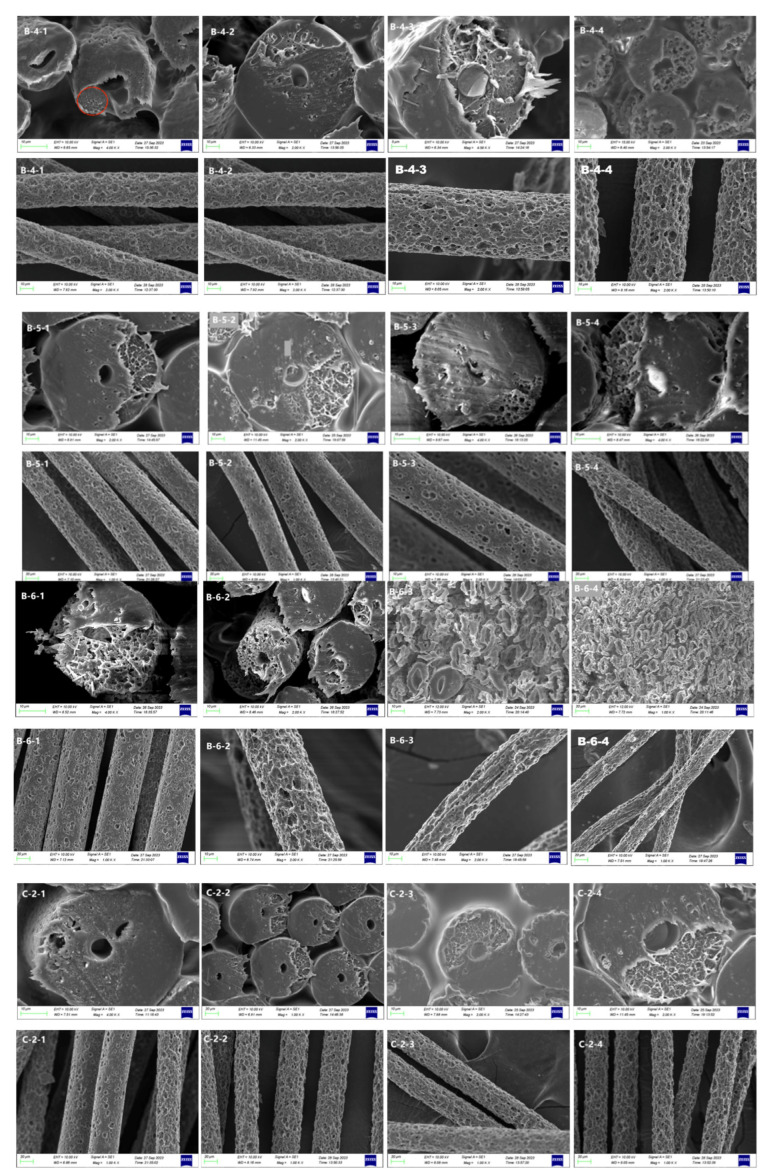
SEM images of transverse and longitudinal sectional morphology before and after acid treatment for samples B and C: upside represent the images of transverse, and downside represent the images of longitudinal. Among them, A-0~A-6 appearing for the first time represents the fiber cross section topography, and A-0~A-6 appearing for the second time represents the fiber longitudinal section topography.

**Figure 8 materials-17-00160-f008:**
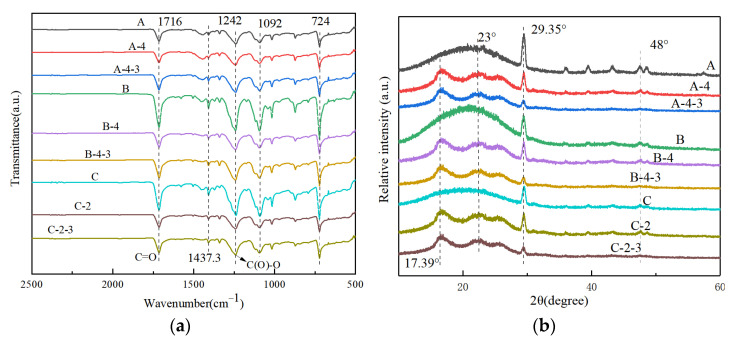
(**a**) FTIR and (**b**) XRD pattern of the samples.

**Figure 9 materials-17-00160-f009:**
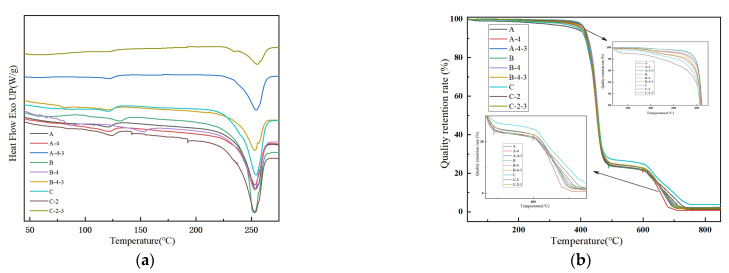
Thermal properties: (**a**) DSC curve; (**b**) TG analysis of fibers before and after post-treatments.

**Figure 10 materials-17-00160-f010:**
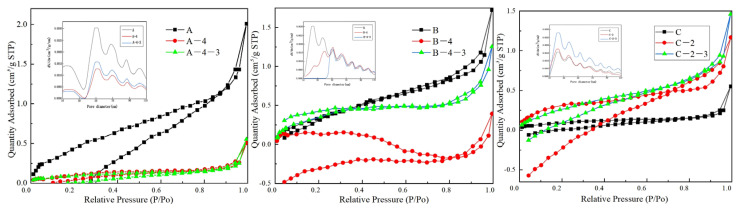
N_2_ adsorption–desorption isotherm and pore size distribution of the sample.

**Figure 11 materials-17-00160-f011:**
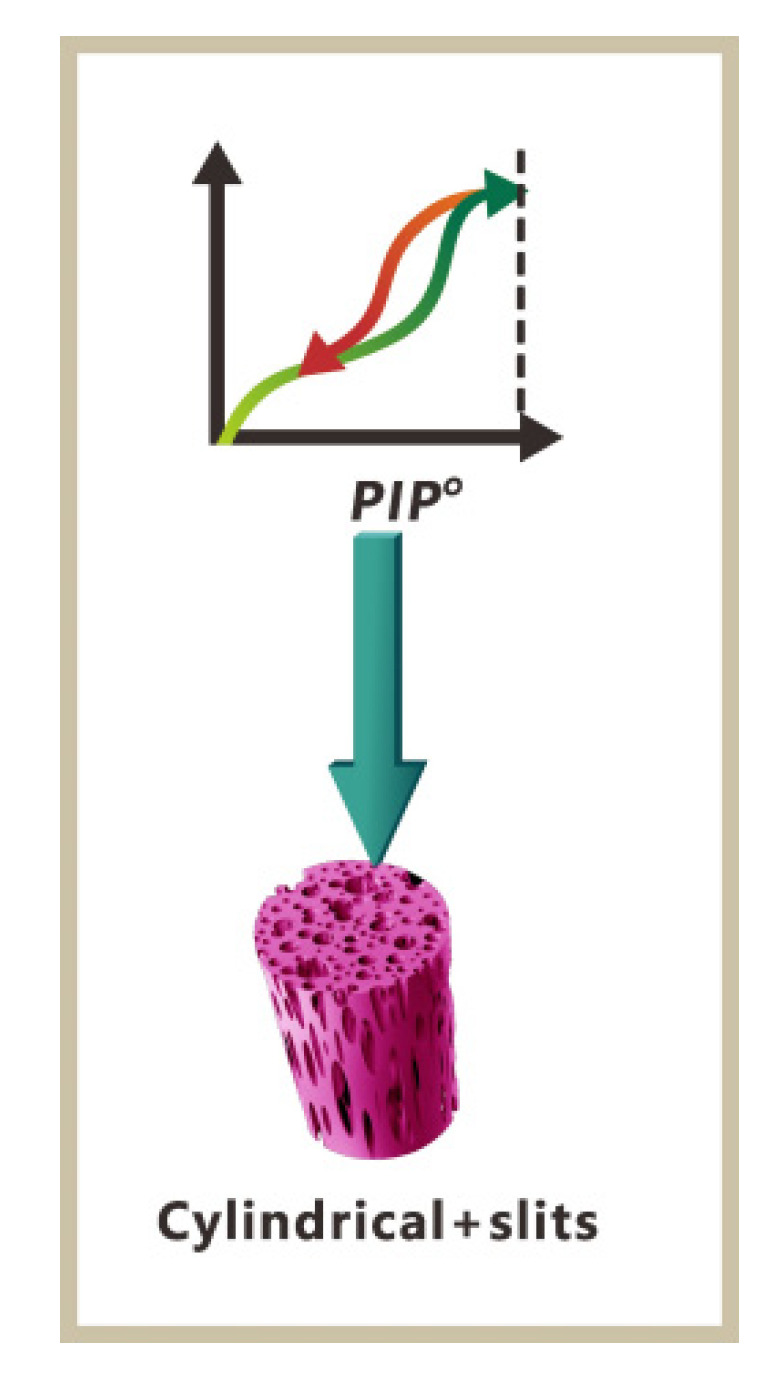
Relationship between orifice hysteresis and orifice profile.

**Table 1 materials-17-00160-t001:** DSC test data of fibers before and after post-treatment.

Sample	T_m_ (°C)	T_c_ (°C)	ΔH_m_ (J/g)	X_c_ (%)
A	255.94	239.20	55.39	39.56
A-4	255.18	238.19	51.42	36.73
A-4-3	255.08	238.87	40.70	29.07
B	253.29	240.01	64.19	41.85
B-4	253.86	237.78	61.68	40.06
B-4-3	253.91	238.87	38.34	27.39
C	252.85	239.09	73.67	42.62
C-2	252.14	238.18	70.71	40.50
C-2-3	252.41	239.84	21.66	15.47

**Table 2 materials-17-00160-t002:** Data of fiber monofilament density and strength after alkali treatment.

Yarn Name	Linear Density/dtex	Breaking Strength/cN/dtex	Elongation at Break/%
A-0	4.91	2.24	232
A-1	20.66	0.42	209
A-2	19.57	0.42	152
A-3	16.87	0.38	117
A-4	14.64	0.31	101
A-5	12.32	0.30	98
A-6	9.95	0.19	73
B-0	5.05	1.84	198
B-1	23.03	0.21	137
B-2	21.63	0.20	109
B-3	19.07	0.19	99
B-4	16.94	0.18	91
B-5	15.75	0.16	49
B-6	6.95	0.15	34
C-0	5.23	1.85	110
C-1	24.56	0.14	89
C-2	20.37	0.14	72
C-3	9.88	0.11	20
C-4	5.81	0.05	16
C-5	0.72	-	-
C-6	-	-	-

**Table 3 materials-17-00160-t003:** Density and strength of fibers after acid treatment.

Yarn Name	Linear Density/dtex	Breaking Strength/cN/dtex	Elongation at Break/%
A-1-1	19.37	0.40	201
A-1-2	18.95	0.38	193
A-1-3	17.97	0.36	181
A-1-4	16.44	0.36	141
A-2-1	18.69	0.36	155
A-2-2	17.83	0.32	122
A-2-3	16.81	0.30	98
A-2-4	15.95	0.28	85
A-3-1	16.21	0.36	105
A-3-2	15.78	0.33	97
A-3-3	14.69	0.32	83
A-3-4	13.17	0.30	56
A-4-1	13.87	0.30	93
A-4-2	12.42	0.26	59
A-4-3	11.76	0.23	47
A-4-4	10.94	0.19	32
A-5-1	11.94	0.30	92
A-5-2	10.83	0.28	63
A-5-3	10.32	0.28	35
A-5-4	9.57	0.21	21
A-6-1	9.13	0.12	68
A-6-2	8.74	0.09	57
A-6-3	8.02	0.07	31
A-6-4	7.63	0.07	12
B-1-1	22.76	0.18	126
B-1-2	21.37	0.15	109
B-1-3	20.45	0.14	94
B-1-4	18.39	0.13	47
B-2-1	20.01	0.19	89
B-2-2	19.32	0.17	64
B-2-3	18.54	0.15	41
B-2-4	17.31	0.11	21
B-3-1	18.74	0.17	84
B-3-2	18.09	0.16	65
B-3-3	17.31	0.12	37
B-3-4	16.84	0.11	17
B-4-1	12.87	0.24	83
B-4-2	12.03	0.22	62
B-4-3	11.21	0.18	51
B-4-4	10.83	0.17	32
B-5-1	9.37	0.22	45
B-5-2	7.92	0.20	34
B-5-3	7.01	0.15	27
B-5-4	6.34	0.15	15
B-6-1	5.96	0.33	30
B-6-2	5.01	0.28	21
B-6-3	4.82	0.22	20
B-6-4	4.03	0.19	17
C-1-1	22.38	0.14	76
C-1-2	21.08	0.11	62
C-1-3	20.05	0.10	43
C-1-4	19.13	0.05	18
C-2-1	18.76	0.10	61
C-2-2	17.01	0.07	41
C-2-3	16.54	0.06	24
C-2-4	15.13	0.05	19
C-3-1	8.25	0.11	20
C-3-2	7.83	0.09	17
C-3-3	6.92	0.08	15
C-3-4	5.73	0.08	9
C-4-1	5.02	0.06	14
C-4-2	4.71	0.05	12
C-4-3	2.02	0.05	8
C-4-4	1.09	0.07	5
C-5-1	-	-	-
C-5-2	-	-	-
C-5-3	-	-	-
C-5-4	-	-	-
C-6-1	-	-	-
C-6-2	-	-	-
C-6-3	-	-	-
C-6-4	-	-	-

**Table 4 materials-17-00160-t004:** Specific surface, pore volume, and aperture of fibers.

Sample	BET Specific Surface Area (m^2^/g)	BET Pore Volume (cm^3^/g)	Mean Aperture (nm)
A	0.44	0.004	6.48
A-4	7.00	0.003	11.27
A-4-3	5.52	0.002	10.87
B	0.88	0.005	6.53
B-4	4.50	0.002	7.82
B-4-3	1.52	0.001	2.13
C	0.96	0.004	7.09
C-2	2.30	0.002	7.06
C-2-3	0.41	0.001	1.48

## Data Availability

Data will be made available on request.

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
