# Peer review of "Preparation and Characterization of Calcium Carbonate Masterbatch–Alkali Soluble Polyester/Polyester Porous Fiber via Melt Spinning"

_materials, 2023, doi:10.3390/ma17010160_

Round 1
Reviewer 1 Report
Comments and Suggestions for Authors
The paper is well written by discussing most interesting technical details related to development of novel porous polyester yarn using inorganic additives and water soluble polyester. This development interesting for academic readers and industrial new developments too.
The abstract is clear and concise with statement of problems, main finding, and principle conclusion.
Introduction has been clearly explained previous research related to authors present study focusing existing such developments.
Materials and method used to develop melt spun porous yarns, evaluation methods are clearly stated.
The results are clear, and discussion was made with supported results for agreement.
In conclusion part, this addressed limitations of this study and provided suggestions.
This paper can be accepted after minor comments and clarification on this paper from myside below
1. How did authors mixed COPET and calcium carbonate? what kind of mixer and mixing condition?
2. What do author mean 40 POY in materials and method section?
3. What was spinneret cross section?, if it was 'C' or hollow, mention what was ration between hollow and solid in yarns?
4. Why do you have different linear density between samples A1,2...B1,2...C1 2..., if it was preset machine parameters to 200D?
5. It would be goof if author indicate strength in Tenacity (Cn/dtex or g/denier)
6. Suggestion: It would be interesting if authors check and report wickability/Moisture management/diffusion/ evaporation of yarn in their next studies.
Author Response
Dear Editor,
We are so appreciative of your letter on our manuscript (2745160), entitled Preparation and Characterization of Calcium carbonate masterbatch -Alkali soluble polyester /polyester porous fiber via melt spinning and we are also extremely grateful to the reviewers' comments on our manuscript. We have carefully considered every comment, and made cautious revisions in the manuscript (marked in red) accordingly. And we also have prepare responds to reviewers in detail as shown in following (marked in blue).
Meanwhile, we tried our best to improve the manuscript. These changes will not influence the paper's content and framework but make the paper more complete and logical. And the changes have marked in red in the revised manuscript. We appreciated for Editors/Reviewers' warm work extremely and hope that the correction will meet with approval.
Once again, we wholeheartedly appreciate all the comments made by the reviewers and you. Please feel free to contact us if anything comes up in the review process.
Your sincerely,
Lifang Liu
lifangliu@dhu.edu.cn
December, 12th 2023
Summary
The paper is well written by discussing most interesting technical details related to development of novel porous polyester yarn using inorganic additives and water soluble polyester. This development interesting for academic readers and industrial new developments too. The abstract is clear and concise with statement of problems, main finding, and principle conclusion. Introduction has been clearly explained previous research related to authors present study focusing existing such developments. Materials and method used to develop melt spun porous yarns, evaluation methods are clearly stated. The results are clear, and discussion was made with supported results for agreement. In conclusion part, this addressed limitations of this study and provided suggestions. This paper can be accepted after minor comments and clarification on this paper from myside below.
- How did authors mixed COPET and calcium carbonate? what kind of mixer and mixing condition?
Response:
Thanks for your important suggestions. We have supplemented COPET and calcium carbonate mixing conditions and mixing equipment information(in line 98-99):We first set up gradient heating and drying for the COPET masterbatch, and crush the dried COPET masterbatch in the PQ-180-800 plastic crusher; Further mixing with CaCO3 powder particles and modified polyester wax mju dispersant; Finally, mix the mixed mixture at 80 ℃, 400 V voltage, and 700 r/min using an SHR 20 L high-speed mixer for 4 consecutive times, each time for 8 minutes, until the mixture is uniform.
2. What do author mean 40 POY in materials and method section?
Response:
Thanks for your carefully review. 40 POY means that the melt spinning extruder is a single screw extruder with a screw diameter of 40 mm(in line 82). the prepared fiber is pre-oriented yarn (POY).
3. What was spinneret cross section?, if it was 'C' or hollow, mention what was ration between hollow and solid in yarns?
Response:
Thanks for your important suggestions. The section of spinneret hole refers to the shape of spinneret hole in spinning assembly. The spinneret adopted in this paper is hollow structure: the middle is hollow, each spinneret has 48 hollow structures, the length-diameter ratio of spinneret hole is 2, and the diameter of single spinneret hole is 0.6mm; The ratio of hollow yarn to solid yarn is 24.41%. (in line 82-84).
4. Why do you have different linear density between samples A1,2...B1,2...C1 2..., if it was preset machine parameters to 200D?
Response:
Thanks for your carefully review. The preset spinning fiber is 200D/48f, equivalent to the monofilaments are 4.167D≈4.625dtex, although the three batches of yarn A, B and C are spun out of the same batch and the same machine, but because of the spinning sequence, personnel, the ratio of different so that it has an error of ±0.8dtex, in this error range is acceptable.In addition, because the prepared fiber is POY pre-oriented silk, it means that the fiber is not drawn when it is prepared, and it will shrink when it encounters high temperature. After treatment with different alkali concentrations (such as A-1 representing 1% alkali concentration treatment), the fiber line density decreased with the increase of alkali concentration.
5. It would be goof if author indicate strength in Tenacity (Cn/dtex or g/denier)
Response:
Thanks for your important suggestions. To avoid ambiguity, the strong force was deleted in the revised draft, and the breaking strength was uniformly expressed. (in line 409-410,437-438).
6. Suggestion: It would be interesting if authors check and report wickability/Moisture management/diffusion/ evaporationof yarn in their next studies.
Response:
Thanks for the advice which can greatly helpful for improving the quality of the paper. This paper focuses on the preparation process of porous fiber and the basic physical properties such as strength and aperture, to find out whether the material can be used as the raw material for lightweight thermal insulation materials. In the follow-up thermal insulation material preparation process, the thermal insulation mechanism analysis will focus on the analysis of yarn twistability/humidity management/diffusion/evaporation and other related content.

Reviewer 2 Report
Comments and Suggestions for Authors
The present work describes the production of porous PET/CaCO3 fibers using different batches of PET. Τhe approach is superficial, but several characterization techniques were performed. The biggest drawbacks of this work are the confusing writing, the lack of much information and the poor interpretation of the results. Some comments are presented below
1. Title is confusing. Acronyms are not necessary, and the names given to the materials are not the most appropriate.
2. The names of the authors at the side column are wrong. Probably copy-paste from the authors section bellow the title.
3. Lines 45-50. The expression is poor. Τhe meaning is not comprehensible.
4. Line 69. The name of the dispersant should be mentioned. Later in the text both CTAB and SIPE are mentioned. Which is the role of each one? The COPET mixture is characterized as “self-made” therefore the existence and the role of each component must be crystal clear.
5. Line 70. In which oscillation does the dissipation value (DF) corresponds to?
6. Line 70. The term “PET masterbatch” predisposes for some kind of mixture, but it seems that it’s about pure PET with specific properties. The authors should clarify this.
7. Figure 1. spelling corrections in the Figure (e.g., “porous fiber” and the red correction line under NaOH)
8. Paragraph 2.2.1. Drying and high-speed mixing conditions
9. Paragraph 2.2.2. What does the “bath ratio” mean? What is the vacuum drying temperature? Please mention in the text.
10. Paragraph 2.3.4, porosimetry tests. Authors should mention the sample pretreatment, the gas used for the porosimetry and which part of the fibers used. Was just a wall fraction or a hollow slice? Moreover, which one of all the pore size distribution methods mentioned was used?
11. Paragraph 2.3.5, DSC and TG. What gas was used during DSC tests?
12. Paragraph 2.3.6, FTIR. Were the measurements conducted using the KBr method or something else? What do the authors mean by the term “better than 0.4 cm-1” in line 165?
13. Paragraph 2.3.7, XRD. What step size was used for the measurements?
14. Line 185. The authors mention about “the fiber’s infiltration into the NaOH solution”. Perhaps they mean the opposite?
15. Line 203. Perhaps the authors mean “…NaOH treatment, and the HCl solution”?
16. Line 222. The box in Figure 5 is yellow and not red.
17. The scale bars can not be read in any of the microscopy images.
18. Paragraph 3.4, Thermal analysis. DSC and TGA are poorly discussed. Authors could comment on the glass transition temperature variations in DSC curves, as well as the fact that TGA curves don’t start from 100 % weight loss, the different amount of the remaining char, and possible reactions between HCl and CaCO3. Moreover, the temperatures of 400, 475 and 485 °C that are given as the onset of some transitions don’t correspond to exactly to the curves. It would be better if the authors also used the DTG curves.
19. Paragraph 3.5.2. Stress-strain curves of the measurements should be also included in the results.
20. Paragraph 3.6. Practically, there are no hysteresis loops in the isotherms and most of the measurements are not acceptable. Same goes for the pore size distribution diagrams. Again, the authors should mention which part of the fiber (wall or hollow slice) did they use for the porosimetry measurements.
Author Response
Dear Editor,
We are so appreciative of your letter on our manuscript (2745160), entitled Preparation and Characterization of Calcium carbonate masterbatch -Alkali soluble polyester /polyester porous fiber via melt spinning and we are also extremely grateful to the reviewers' comments on our manuscript. We have carefully considered every comment, and made cautious revisions in the manuscript (marked in red) accordingly. And we also have prepare responds to reviewers in detail as shown in following (marked in blue).
Meanwhile, we tried our best to improve the manuscript. These changes will not influence the paper's content and framework but make the paper more complete and logical. And the changes have marked in red in the revised manuscript. We appreciated for Editors/Reviewers' warm work extremely and hope that the correction will meet with approval.
Once again, we wholeheartedly appreciate all the comments made by the reviewers and you. Please feel free to contact us if anything comes up in the review process.
Your sincerely,
Lifang Liu
lifangliu@dhu.edu.cn
December, 12th 2023
Summary:
The present work describes the production of porous PET/CaCO3 fibers using different batches of PET. Τhe approach is superficial, but several characterization techniques were performed. The biggest drawbacks of this work are the confusing writing, the lack of much information and the poor interpretation of the results. Some comments are presented below.
- Title is confusing. Acronyms are not necessary, and the names given to the materials are not the most appropriate.
Response:
Thanks for your important suggestions. Acronyms have been removed(in line 2-4) . In order to facilitate the recording and differentiation of samples, the sample is simply named A, B, C instead.
- The names of the authors at the side column are wrong. Probably copy-paste from the authors section bellow the title.
Response:
Thanks for your important suggestions. The author's name has been corrected in the sidebar(in line 16-17) .
- Lines 45-50. The expression is poor.Τhe meaning is not comprehensible.
Response:
Thanks for your carefully review. Lines 45-50 have been rewritten(in line 45-51) . By filling CaCO3 to modify traditional fibers such as polyester and polypropylene, both the technical maturity and manufacturing maturity have met the requirements of industrial production, but this technology is now mostly used in the plastic industry, and the content of CaCO3 in industrial production is mostly within the range of 1%-10%, and in the field of textile application is still in the immature stage. In particular, there are few reports on the preparation of fiber grade by melt spinning by mixing calcium carbonate particles and alkali soluble polyester masterbatch granulation.
- Line 69. The name of the dispersant should be mentioned. Later in the text both CTAB and SIPE are mentioned. Which is the role of each one? The COPET mixture is characterized as “self-made” therefore the existence and the role of each component must be crystal clear.
Response:
Thanks for your important suggestions. The COPET mixtures are characterized as "homemade", with the presence and action of each component as follows(in line 71-76, 80-81): COPET/CaCO3 masterbatch (called CTCA masterbatch) (self-made, COPET/CaCO3/modified polyester wax mju dispersant/68.5/30/1.5, [η]=0.332 dL/g, Filter press value 5.01 MPa; among them, COPET is made by adding modified components to ordinary PET synthesis, including SIPE, EG and other components and SIPE monomer is mainly used in the synthesis of polyester, which is beneficial to improve the solubility and dyeing property of polyester.); Cetyltrimethyl ammonium bromide (CTAB,96%) is a penetrant and surfactant.
- What was spinneret cross section?, if it was 'C' or hollow, mention what was ration between hollow and solid in yarns?
Response:
Thanks for your important suggestions. The section of spinneret hole refers to the shape of spinneret hole in spinning assembly. The spinneret adopted in this paper is hollow structure: the middle is hollow, each spinneret has 48 hollow structures, the length-diameter ratio of spinneret hole is 2, and the diameter of single spinneret hole is 0.6mm and the ratio of hollow in yarn is 24.41%(in line 82-84).
- Line 70. In which oscillation does the dissipation value (DF) corresponds to?
Response:
Thanks for your important suggestions. The DF value is filter press value and to mix the color masterbatch and carrier resin according to a certain proportion, melt and extrude with a single hole extruder, measure the change value of the head pressure over time, and use the increase value of the head pressure to judge the degree of fine pigment contained in the color masterbatch.
- Line 70. The term “PET masterbatch” predisposes for some kind of mixture, but it seems that it’s about pure PET with specific properties. The authors should clarify this.
Response:
Thanks for your important suggestions. “PET masterbatch” change to PET slice(in line 76, 97).
- Figure 1. spelling corrections in the Figure (e.g., “porous fiber”and the red correction line under NaOH)
Response:
Thanks for your important suggestions. Figure 1(in line 92-93). “pore fiber” change to “porous fiber” and the red correction line under NaOH was removed.
- Paragraph 2.2.1. Drying and high-speed mixing conditions
Response:
Thanks for your important suggestions. We first set up gradient heating and drying for the CTCA masterbatch: CTCA masterbatch was first dried in a vacuum oven at 80 ℃ for 4 h, and then dried at 110 ℃ for 12 h. PET powder particles were dried in a vacuum drying oven at 110 ℃ for 5 h; After thorough drying, the self-made COPET/CaCO3 masterbatch, denoted as CTCA masterbatch, and PET slice were blended at different ratios (95/5 as A, 85/15 as B, and 80/20 as C) were mixed in a high-speed mixer at 80℃, 400V voltage, 700r/min for 4 consecutive times, each time for 8 minutes.(in line 95-101)
- Paragraph 2.2.2. What does the “bath ratio” mean? What is the vacuum drying temperature? Please mention in the text.
Response:
Thanks for your important suggestions. Bath ratio refers to the weight ratio of the fiber to the solution; they are dried at a vacuum drying temperature of 100 ℃ for 2 h.(in line 114-115)
- Paragraph 2.3.4, porosimetry tests. Authors should mention the sample pretreatment, the gas used for the porosimetry and which part of the fibers used. Was just a wall fraction or a hollow slice? Moreover, which one of all the pore size distribution methods mentioned was used?
Response:
Thanks for your important suggestions. The untreated and treated fibers were randomly cut to 0.1 g each for testing, Combined with BET and BJH model principles, the specific surface area and pore size distribution were analyzed respectively. The test conditions were N2 adsorption, the analysis tank temperature was -195.85 ℃, the low pressure dose was 6.0 cm3/g, the equilibrium interval was 15 s, the degassation temperature was 100 ℃, and the degassation time was 6 h.(in line 161-166)
- Paragraph 2.3.5, DSC and TG. What gas was used during DSC tests?
Response:
Thanks for your important suggestions. The gas used in DSC testing is nitrogen.(in line 171).
- Paragraph 2.3.6, FTIR. Were the measurements conducted using the KBr method or something else? What do the authors mean by the term “better than 0.4 cm-1” in line 165?
Response:
Thanks for your important suggestions. the composition and structure of the sample were analyzed according to the photoacoustic spectrometry method specified in GB/T 6040-2019 "General Rules for Infrared Spectral Analysis Methods"(in line 179-181); Better than 0.4 cm-1 means that the nearest distance between the two peaks can be distinguished by 0.4.
- Paragraph 2.3.7, XRD. What step size was used for the measurements?
Response:
Thanks for your important suggestions. the step width is 0.02° and a scanning speed of 0.15 s /step.(in line 188)
- Line 185. The authors mention about “the fiber’s infiltration into the NaOH solution”. Perhaps they mean the opposite?
Response:
Thanks for your important suggestions. After careful consideration, it has now been changed to “CTAB, a series of quaternary ammonium surfactants, plays a significant role in reducing the interfacial tension between PET and NaOH aqueous solution. Thus, the NaOH solution is more easily immersed in the fiber interior, which is conducive to hydrolysis reaction”.(in line 202-204)
- Line 203. Perhaps the authors mean “…NaOH treatment, and the HCl solution”?
Response:
Thanks for your important suggestions. We revised as: when the HCl concentration and the amount of CTCA masterbatch remain constant, the acid decrement increases as the NaOH base concentration rises.(in line 218-220)
- Line 222. The box in Figure 5 is yellow and not red.
Response:
Thanks for your important suggestions. The box in Figure 5 is yellow(in line 241).
- The scale bars can not be read in any of the microscopy images.
Response:
Thanks for your important suggestions. The scale in the SEM image is the scale automatically recorded by the test instrument according to the observation multiple used in the test.
- Paragraph 3.4, Thermal analysis. DSC and TGA are poorly discussed. Authors could comment on the glass transition temperature variations in DSC curves, as well as the fact that TGA curves don’t start from 100 % weight loss, the different amount of the remaining char, and possible reactions between HCl and CaCO3. Moreover, the temperatures of 400, 475 and 485°C that are given as the onset of some transitions don’t correspond to exactly to the curves. It would be better if the authors also used the DTG curves.
Response:
Thanks for your important suggestions. For TG test, we re-conducted the test and result analysis(in line 384-413):The thermal stability analysis of the samples in Figure 8b reveals a consistent thermal degradation behavior among all samples. The thermal decomposition temperature, approximately 390 ℃, aligns with the thermal decomposition temperature of pure PET [32,14]. The thermal decomposition of the sample is divided into two stages. The first stage is about 30 ℃~600 ℃, which is mainly COPET/PET resin matrix decomposition. The second stage is about 600 ℃~800 ℃, which is the decomposition of CaCO3 inorganic particles [39,40]. In the first stage, the thermal decomposition temperature is about 390 ℃, which is basically close to the thermal decomposition temperature of pure PET polyester [34,35]. For both untreated and alkali-treated samples, with the increase of CTCA masterbatch content, the thermal decomposition temperature of the samples first increased and then decreased. For the acid-treated samples, the thermal decomposition temperature decreased with the increase of CTCA masterbatch content. In the second stage, for untreated and alkali treated samples, the carbon residue increased with the increase of CTCA masterbatch content. After acid treatment, with the increase of CTCA masterbatch content, the carbon residue increased first and then decreased. This is because the content of CTCA masterbatch increases, the content of CaCO3 inorganic particles in the fiber increases, and the CaCO3 particles start to decompose at about 600 ℃~800 ℃. Therefore, the addition of CaCO3 at this stage can increase the thermal decomposition temperature of the fiber, and also increase the quality of carbon residue. When treated with alkali, OH- reacts with COPET and PET on the fiber surface step by step, and finally forms pores on the fiber surface and inside. When the pore volume is small, it is helpful to increase the thermal decomposition temperature and carbon residue. However, when there are too many pores, CaCO3 particles adsorbed on the fiber will be detached from the fiber, and the fiber matrix itself will be hydrolyzed and consumed, so the thermal decomposition temperature and carbon residue will decrease. When the acid solution reacts with CaCO3 inorganic particles at the beginning, the reaction rate is slow. With the increase of CaCO3 particles, the amount of carbon residue increases, and with the increase of CTCA masterbatch content, the reaction rate with acid is accelerated, the consumption rate is greater than the addition rate of CaCO3 particles, and the gas generated by the reaction dissolves into pores, resulting in more pores on the fiber surface. The probability of CaCO3 particles separating from fiber is further increased, so the thermal decomposition temperature of fiber and the amount of carbon residue are reduced.
- Paragraph 3.5.2. Stress-strain curves of the measurements should be also included in the results.
Response:
Thanks for your important suggestions. There are too many samples involved and the stress-strain curve is cluttered when put into a single graph. The following figure is a partial stress-strain curve, and it is not recommended to include a stress-strain curve.
- Paragraph 3.6. Practically, there are no hysteresis loops in the isotherms and most of the measurements are not acceptable. Same goes for the pore size distribution diagrams. Again, the authors should mention which part of the fiber (wall or hollow slice) did they use for the porosimetry measurements.
Response:
Thanks for your important suggestions. The untreated and treated fibers were randomly cut to 0.1 g each for testing.(in line 161)
We retested the test and the results were the same as those in the text. However, after discussion and consultation, based on the principle of image testing, we write a program to characterize and analyze the porosity. The following are our procedures and test results, please ask experts to help see if it can be replaced with this method.
Sample B-4-3 was selected and the porosity was calculated by Matlab software.
Figure (a) shows the SEM diagram of the B-4-3 optical fiber cross section. (b) is the section image processed by Matlab software, and the total area is set to be 100%. According to the calculation results, the pore area (the pixel value of the black part) accounts for 53.94% of the total area, that is, the porosity of the fiber section is 53.94%.
(a) (b)
Matlab software calculation code:
clc;
clear;
img=imread('mm.png');
grayimg=rgb2gray(img);
BWimg=grayimg;
[width,height]=size(grayimg);
figure;
imshow(grayimg);
T1=245;
T2=140;
ii=0;
jj=0;
for i=1:width
for j=1:height
if(grayimg(i,j)<T1)
BWimg(i,j)=0;
ii=ii+1;
else
BWimg(i,j)=250;
jj=jj+1
end
end
end
figure;
imshow(BWimg);
a=ii/(ii+jj)
ii=0;
jj=0;
for i=1:width
for j=1:height
if(grayimg(i,j)<T2)
BWimg(i,j)=0;
ii=ii+1;
else
BWimg(i,j)=250;
jj=jj+1
end
end
end
figure;
imshow(BWimg);
b=ii/(ii+jj)
v=b/a

Reviewer 3 Report
Comments and Suggestions for Authors
(87-88) What dictated such ratios of CTCA/PET masterbatches?
(line 97) What does this bath ratio refer to?
(line 109) From the description posted previously (lines 69 and 86-88) it resulted that CACT was 95 units and PET was 5 units (i.e. vice versa)
(137-140) The capabilities of the microscope are given here, not the conditions in which the tests were performer
(227-235 and 263-270) why were the photos taken at different magnifications? Does this mean that fibers with different cross-sections were obtained or these cross-sections were changed as a result of alkali treatment?
It is difficult to determine which photos show longitudinal sections of the fibers and which show the longitudinal appearance of the surface of these fibers.
Why do fibers have lumen in addition to pores?
(236-248) Perhaps the reaction equation would make this description more familiar to the readers?
(287-296) What fiber structure did the authors want to achieve in their experiments?
Authors should consider the content of the data in Table 3. In this form, the table resembles a report from laboratory tests, rather than the data that is relevant and discussed in the article.
(428-432) Could the authors describe more clearly which attempts they find satisfactory and why, and which ones do not meet their expectations?
(line 458) What technologies do the authors refer to in this comparison? What percentage of fiber porosity was achieved using the presented method?
Author Response
Dear Editor,
We are so appreciative of your letter on our manuscript (2745160), entitled Preparation and Characterization of Calcium carbonate masterbatch -Alkali soluble polyester /polyester porous fiber via melt spinning and we are also extremely grateful to the reviewers' comments on our manuscript. We have carefully considered every comment, and made cautious revisions in the manuscript (marked in red) accordingly. And we also have prepare responds to reviewers in detail as shown in following (marked in blue).
Meanwhile, we tried our best to improve the manuscript. These changes will not influence the paper's content and framework but make the paper more complete and logical. And the changes have marked in red in the revised manuscript. We appreciated for Editors/Reviewers' warm work extremely and hope that the correction will meet with approval.
Once again, we wholeheartedly appreciate all the comments made by the reviewers and you. Please feel free to contact us if anything comes up in the review process.
Your sincerely,
Lifang Liu
lifangliu@dhu.edu.cn
December, 12th 2023
Some comments are presented below.
- (87-88) What dictated such ratios ofCTCA/PET masterbatches?
Response:
Thanks for your important suggestions. The author first conducted a spinning pre-experiment, starting with a high proportion and continuing until continuous spinning can occur at the spinnery mouth(see figure), with the highest proportion when the particles are evenly dispersed. Then, under this proportion, experimental schemes of different proportions were designed to verify the dispersion of inorganic particles in the fibers and the influence on the properties of the fibers. Based on the preliminary test results, three ratios of 95/5, 85/15, 80/20 were selected in this paper.
- (line 97) What does this bath ratio refer to?
Response:
Thanks for your important suggestions. Bath ratio refers to the weight ratio of the fiber to the solution.
- (line 109) From the description posted previously (lines 69 and 86-88) it resulted that CACT was 95units and PET was 5 units (i.e. vice versa)
Response:
Thanks for your important suggestions. Represents a named method, For example, A-1-1 represents A fiber with a CACT/PET ratio of 95/5, treated with a solution with a base mass fraction of 1% and an acid concentration of 1mol/L.
- (137-140) The capabilities of the microscope are given here, not the conditions in which the tests were performer
Response:
Thanks for your important suggestions. Now 137-140 has been changed to the following(in line 146-154):
The transverse and longitudinal morphology of samples were observed using the EVO 10 Zeiss tungsten filament scanning electron microscope (Carl Zeiss Optics (China) Co., LTD.), and the accelerated voltage was 0.2kV ~ 30kV. During sample preparation, the cross section was cut by Y172 fiber slicer according to GB/T 10685-2007 "Wool Fiber Diameter Test Method projection microscope Method". The longitudinal section was subjected to liquid nitrogen embritching method. The sample was frozen in liquid nitrogen for 20 min, and then embritched with scissors frozen in liquid nitrogen. The sample was vertically adhered to the electron microscope table, and the sample was sprayed with platinum for 150 s at a current of 15 mA before testing.
- (227-235 and 263-270) why were the photos taken at different magnifications? Does this mean that fibers with different cross-sections were obtained or these cross-sections were changed as a result of alkali treatment?
Response:
Thanks for your important suggestions. All SEM images of fibers have been changed to images with the same multiples.(in line 247-250,283-297).
- It is difficult to determine which photos show longitudinal sections of the fibers and which show the longitudinal appearance of the surface of these fibers.
Response:
Thanks for your important suggestions. The same number contains two pictures. The picture that appears the number for the first time is a cross section picture, and the picture that appears the number for the second time in the following row is a longitudinal surface picture.
- Why do fibers have lumen in addition to pores?
Response:
Thanks for your important suggestions. When spinning, the spinning assembly is a hollow spinneret. The spinneret adopted in this paper is hollow structure: the middle is hollow, each spinneret has 48 hollow structures, the length-diameter ratio of spinneret hole is 2, and the diameter of single spinneret hole is 0.6mm; The ratio of hollow yarn to solid yarn is 24.41%.(in line 82-84) .
- (236-248) Perhaps the reaction equation would make this description more familiar to the readers?
Response:
Thanks for your important suggestions. The image is as follows(in line 268):
- (287-296) What fiber structure did the authors want to achieve in their experiments?
Response:
Thanks for your carefully review. To obtain this fiber structure, see figure:
- Authors should consider the content of the data in Table 3. In this form, the table resembles a report from laboratory tests, rather than the data that is relevant and discussed in the article.
Response:
Thanks for your carefully review. In this paper, 93 groups of orthogonal tests are done, and the strength tests of 93 groups of fibers are carried out, and the data are summarized. If it is displayed graphically, the overall appearance is cluttered due to the large amount of data (see figure,Partial data), and the table is the most clear way of expression.
- (428-432) Could the authors describe more clearly which attempts they find satisfactory and why, and which ones do not meet their expectations?
Response:
Thanks for your carefully review. The nitrogen adsorption desorption curve reflects the size and pore size distribution of the inner pores of the fibers. The result we want is that the hysteresis curves of N2 adsorption-desorption curves of B series samples have the same width and high pore size uniformity.
- (line 458) What technologies do the authors refer to in this comparison? What percentage of fiber porosity was achieved using the presented method?
Response:
Thanks for your carefully review. In this paper, BET and BJH model principle and IUPAC classification method are used to compare and analyze fiber specific surface area and pore size distribution.
based on the principle of image testing, we write a program to characterize and analyze the porosity. The following are our procedures and test results, please ask experts to help see if it can be replaced with this method.
Sample B-4-3 was selected and the porosity was calculated by Matlab software.
Figure (a) shows the SEM diagram of the B-4-3 optical fiber cross section. (b) is the section image processed by Matlab software, and the total area is set to be 100%. According to the calculation results, the pore area (the pixel value of the black part) accounts for 53.94% of the total area, that is, the porosity of the fiber section is 53.94%.
(a) (b)
Matlab software calculation code:
clc;
clear;
img=imread('mm.png');
grayimg=rgb2gray(img);
BWimg=grayimg;
[width,height]=size(grayimg);
figure;
imshow(grayimg);
T1=245;
T2=140;
ii=0;
jj=0;
for i=1:width
for j=1:height
if(grayimg(i,j)<T1)
BWimg(i,j)=0;
ii=ii+1;
else
BWimg(i,j)=250;
jj=jj+1
end
end
end
figure;
imshow(BWimg);
a=ii/(ii+jj)
ii=0;
jj=0;
for i=1:width
for j=1:height
if(grayimg(i,j)<T2)
BWimg(i,j)=0;
ii=ii+1;
else
BWimg(i,j)=250;
jj=jj+1
end
end
end
figure;
imshow(BWimg);
b=ii/(ii+jj)
v=b/a

Round 2
Reviewer 2 Report
Comments and Suggestions for Authors
Dear authors,
thank you for your effort and time spent to address all the comments.
Your work is accepted in the present form
Author Response
Dear reviewer,
We are so appreciative of your comments on our manuscript (2745160), entitled Preparation and Characterization of Calcium carbonate masterbatch -Alkali soluble polyester /polyester porous fiber via melt spinning and thanks for the advice which can greatly helpful for improving the quality of the paper. We appreciated for Reviewer' warm work extremely .
Once again, we wholeheartedly appreciate all the comments made by the reviewer.
Your sincerely,
Lifang Liu
lifangliu@dhu.edu.cn
December, 22th 2023

Reviewer 3 Report
Comments and Suggestions for Authors
I would like to thank the Authors for their responses to my comments and for the corrections made to the manuscript.
Author Response

(The authors gave the same response as above.)
